# Selection, Characterization, and Optimization of DNA Aptamers against Challenging Marine Biotoxin Gymnodimine-A for Biosensing Application

**DOI:** 10.3390/toxins14030195

**Published:** 2022-03-05

**Authors:** Xiaojuan Zhang, Yun Gao, Bowen Deng, Bo Hu, Luming Zhao, Han Guo, Chengfang Yang, Zhenxia Ma, Mingjuan Sun, Binghua Jiao, Lianghua Wang

**Affiliations:** 1Department of Biochemistry and Molecular Biology, College of Basic Medical Sciences, Navy Medical University, Shanghai 200433, China; emilyzhangxj@126.com (X.Z.); gaoyun2014@sohu.com (Y.G.); dengbwgsy9373@gmail.com (B.D.); zlm19960726@163.com (L.Z.); ggatecnu@163.com (H.G.); nicole20220104@126.com (C.Y.); mazhenxia0515@sina.com (Z.M.); sunmj@smmu.edu.cn (M.S.); 2College of Medicine, Shaoxing University, 900th Chengnan Avenue, Shaoxing 312000, China; 3Department of Marine Biomedicine and Polar Medicine, Naval Medical Center of PLA, Navy Medical University, Shanghai 200433, China; hb8601@163.com

**Keywords:** gymnodimine-A, aptamer, aptasensor, biolayer interferometry

## Abstract

Gymnodimines (GYMs), belonging to cyclic imines (CIs), are characterized as fast-acting toxins, and may pose potential risks to human health and the aquaculture industry through the contamination of sea food. The existing detection methods of GYMs have certain defects in practice, such as ethical problems or the requirement of complicated equipment. As novel molecular recognition elements, aptamers have been applied in many areas, including the detection of marine biotoxins. However, GYMs are liposoluble molecules with low molecular weight and limited numbers of chemical groups, which are considered as “challenging” targets for aptamers selection. In this study, Capture-SELEX was used as the main strategy in screening aptamers targeting gymnodimine-A (GYM-A), and an aptamer named G48nop, with the highest *K*_D_ value of 95.30 nM, was successfully obtained by screening and optimization. G48nop showed high specificity towards GYM-A. Based on this, a novel aptasensor based on biolayer interferometry (BLI) technology was established in detecting GYM-A. This aptasensor showed a detection range from 55 to 1400 nM (linear range from 55 to 875 nM) and a limit of detection (LOD) of 6.21 nM. Spiking experiments in real samples indicated the recovery rate of this aptasensor, ranging from 96.65% to 109.67%. This is the first study to report an aptamer with high affinity and specificity for the challenging marine biotoxin GYM-A, and the new established aptasensor may be used as a reliable and efficient tool for the detection and monitoring of GYMs in the future.

## 1. Introduction

Gymnodimines (GYMs) are the smallest members of cyclic imines (CIs) in terms of molecular weight. They are characterized as fast-acting toxins, which can accumulate in filter feeding shellfish and may pose a potential risk to human health and the aquaculture industry [1,2,3]. Being first discovered in New Zealand oysters [4], GYMs have been detected in pipis (*Donax deltoides*), mussels (*Modiolus proclivis*), oysters (*Saccostrea glomerata*), and seawater in many countries [1,5,6]. GYMs can also be isolated from the dinoflagellates *Alexandrium ostenfeldii* [7] (*Alexandrium peruvianum* [8,9]) and *Karenia selliformis* [10], which are harmful algal bloom species [11,12,13,14]. The stereochemical structure of gymnodimine-A (GYM-A) was first elucidated by nuclear magnetic resonance (NMR) in 1995 [4] and confirmed by X-ray diffraction in 1997 [15]. Then, novel GYMs and their congeners were isolated successively, including 12-methyl GYM-A [16], GYM-B [17], 12-methyl GYM-B [18], GYM-C [19], GYM-D [20], 16-desmethyl GYM-D, and GYM-E [7]. Their respective molecular weight is around 520 (Figure 1). As liposoluble toxins, the typical chemical structure of GYMs contains a macrocycle ring, a spirocyclic imine ring, and an ether moiety.

Intracerebroventricular injection (i.c.) or intraperitoneal injection (i.p.) of GYM-A or GYM-B caused rapid death in mice. The average LD_50_ of GYM-A were 3, 80, and 100 μg/kg for i.c., i.p., and subcutaneous injection, respectively [21]. Poisoning manifestations included hyperactivity, jumping, paralysis of the hind legs, and severe dyspnea. These results indicated that GYM-A could act on the central nervous system and peripheral nervous system [21]. GYM-A was also shown to bind to muscular and neuronal nicotinic acetylcholine receptors (nAChR) with high affinity, in a reversible way, in vitro [21,22,23,24]. In addition, GYMs might promote cells to be more susceptible to the toxic effects caused by other algal toxins as Neuro2a cells become more sensitive to okadaic acid (OA) by pre-exposure to GYM-A and its analogues [25].

No incidents related to human GYM poisoning cases have been reported, and GYMs in shellfish are not regulated in the European Union (EU) or other authoritative organizations [26,27,28]. Given their “fast-acting” toxicity, the potential threat of these toxins to human health and the aquaculture industry remains to be considered [29,30]. Because of GYMs, a prophylactic closure of conchylicultural activities in Foveaux Strait (New Zealand) was forced by mouse bioassay mortality following intraperitoneal injection with contaminated shellfish [31]. A request has already been proposed by UK food safety authorities to examine these toxins more closely for effective management [29]. In addition, the toxicological information of GYMs, including acute and chronic toxicity, still needs to be further investigated [32]. Moreover, the presence of GYMs might interfere with the detection of other liposoluble marine biotoxins and lead to false positive results [33]. Therefore, research work to establish effective detection and monitoring methods for GYMs is of great value.

The existing detection methods of GYMs, including mouse bioassay (MBA), liquid chromatography–tandem mass spectrometry (LC–MS/MS), and receptor-binding assay (RBA), have certain defects in practice. MBA is a traditional way to detect GYMs in seafood [21,34], but it lacks detection specificity and involves animal ethics [27]. LC–MS/MS has been laid down by the European Union as a reference method for identification of lipophilic marine toxins (LMTs) [35]. Various LC–MS/MS methods have been developed by the advancement of technology. For example, the establishment of the liquid chromatography–high-resolution mass spectrometry (LC–HRMS) approach is conducive to the discovery of new derivatives of GYMs and the in-depth investigation of the metabolic characteristics of shellfish samples [36]. The combination of the dispersive liquid–liquid microextraction (DLLME) with the liquid chromatography with triple quadrupole mass spectrometry (LC–QqQ-MS/MS) allowed for the sensitive and selective analysis of thirteen lipophilic marine toxins, including GYM in seawater samples [37]. Although the performance of LC–MS/MS methods, including specificity, sample consumption, detection time, or the low limit of detection (LOD), continues to improve, these methods are still expensive, complex, time-consuming, and need special personnel, sophisticated instrument, or certified toxin of various congeners of GYMs [28,38]. RBA methods were based on the fact that GYM-A can replace α-bungarotoxin on nAChRs by competitive binding, which are abundant in the electric organs of *Torpedo californica* [39] and *T. marmorata* [40]. Thus, GYM-A can be detected by labeling α-bungarotoxin with fluorescence or biotin. Although results can be obtained very quickly in RBA, poor specificity and difficulties in obtaining nAChRs are the main drawbacks of these methods [41].

Aptamers, which are single stranded DNA or RNA oligonucleotides, are perceived as chemical antibodies. They have the ability to bind to specific targets by crimping and folding into specific tertiary structures. Aptamers are screened from DNA or RNA libraries though systematic evolution of ligands by exponential enrichment (SELEX) [42,43]. As novel recognition element, aptamers have many advantages [44,45]. Firstly, the affinity and selectivity of aptamers are comparable to those of antibodies. Secondly, no animals are involved in the aptamer development process. Small molecules, with toxicity and non-immunogenicity, can also be screened to obtain high-affinity aptamers. Thirdly, aptamers have good thermal and chemical stability and are easy to store and transport. The denatured aptamer can be renatured under suitable conditions, enabling it to be used repeatedly as the molecular recognition element of biosensors. Fourthly, the aptamer is easily labeled and chemically modified, and it can produce different structures when binding to its targets, making it easy to develop into different types of biosensors [44,45]. At present, aptamers have good application prospects in many fields [44,45,46]. The high affinity and high specificity of the aptamer make it an ideal diagnostic reagent, which has the potential to replace antibodies in pathogen recognition, cancer diagnosis, detection of environmental pollutants, and food safety [47,48,49,50]. Aptamers can also serve as ideal therapeutic agents due to competition with specific small molecules or protein ligands [51,52]. In addition, aptamers can serve as carriers to deliver many therapeutic agents to specific tissues or cells [53,54].

Marine toxins, which are mainly small molecular compounds, are one of the main factors leading to marine food poisoning [44]. Currently, high-affinity and high-specificity aptamers have been successfully obtained for some representative marine toxins, including saxitoxin (STX) [55], OA [56], palytoxin (PLTX) [57], gonyautoxin 1/4 (GTX 1/4) [58], brevetoxin-2 (BTX-2) [59], and microcystin-LR (MC-LR) [60], etc. Most of them have been developed into specific aptasensors [56,57,58,59,60,61]. These aptasensors, with high specificity, high sensitivity, convenience, time saving, and low cost, are ideal candidates for the detection of marine toxins [44,56,57,58,61]. As typical marine toxins, GYMs are lipid-soluble and have limited numbers of chemical groups, which are considered to be “challenging” molecules for aptamer screening [62,63]. So far, no GYMs-related aptamers have been reported. However, with the continuous improvement of screening technology, there have been some reports on obtaining high-affinity and high-specificity aptamers for highly lipid-soluble targets, such as steroids [62] and cannabinoids [63]. In this research, aptamers for challenging small molecule GYM-A were selected by Capture-SELEX for the first time, and an aptamer with high affinity and specificity was successfully obtained by screening and optimization. This aptamer was further developed into a biolayer interferometry (BLI)-based aptasensor. This aptasensor has high specificity, high sensitivity, and an ideal recovery rate in real samples for the detection of GYM-A. Thus, this aptasensor may be used as an efficient and sensitive tool in detecting and monitoring of GYM-A in the future.

## 2. Results and Discussion

### 2.1. GYM-A Aptamer Selection by Capture-SELEX

GYM-A is a challenging target for screening of aptamers as this molecule is liposoluble and contains a limited number of chemical groups, which makes it hard to be modified and immobilized onto solid medium. Besides, the immobilization of GYM-A on solid medium may cause the change in its native structure, which may reduce the success rate of screening. Therefore, Capture-SELEX was used as the main screening method in this study. In this strategy, ssDNA libraries are immobilized onto magnetic beads by complementary base pairing, while GYM-A that contains native structure is in the solution [64,65]. The workflow of Capture-SELEX in this research is shown in Figure 2.

The design of the library, primers, and capture oligo referred to the report by Reinemann [66] (Appendix A). In this research, the longer random sequence of the library was shortened to 30 nt according to the original method [66], so the total length of the library was 88 nt. About 10^15^ ssDNA oligonucleotides were contained in the first library. Polypeptide marine biotoxin microcystin-LR (MC-LR), whose structure is rich in several chemical groups and does not contain spirocyclic imine, was used as the counter target in round 6–12. As shown in Appendix A and Appendix A, the selection pressure increased by adding the counter target of MC-LR, increasing the amount and incubation time of MC-LR, and reducing the incubation time of GYM-A. The increased pressure are beneficial to obtain potential aptamers with high affinity and specificity. During the screening process, the recovery rate continued to increase until round 12, so the iterative selection was stopped (Appendix A). Therefore, the enriched ssDNA of round 12 was cloned and sequenced for further analysis.

### 2.2. Cloning, Sequencing, Selection, and Affinity Identification of Candidate Aptamers

The enriched ssDNA from round 12 was cloned into pGEM-18T vector and a total of 80 clones were selected randomly and sequenced. The obtained sequences were analyzed by Clustal X 2.0 software [67] and divided into six families according to their sequence similarities. The secondary structure and Gibbs free energy of these sequences were predicted by the web server of mfold [68]. Finally, the six sequences with the highest homology, or the lowest Gibbs free energy, were chosen from each family as candidate aptamers for further identification. The affinity of these candidate aptamers was measured by BLI assays (Table 1). The aptamer G48 (Figure 3A) with the lowest *K*_D_ value was chosen for further optimization.

### 2.3. Truncation of Aptamer G48

It has been reported that the affinity of several aptamers was optimized by sequence truncation [58,69,70]. Therefore, we generated some truncated forms of G48 and measured the affinity of them by BLI (Table 2). Firstly, we generated G48nop (Figure 3B) by removing the primer binding sequences in G48. The resulting *K*_D_ value of G48nop (95.30 nM) was about one third of that of G48 (288 nM), which indicated that the primer binding sequences of G48 were not beneficial for the interaction between G48 and its target. Then, we tried to improve the affinity of G48nop by the removal of some stem sequences in the stem-loop structure, based on the predicted secondary structure of G48nop. However, the affinities of the generated G48nors and G48norsj were not improved accordingly (Table 2, Appendix A). These results imply that the stem in loop-stem structure may have a certain effect for the binding.

A total of 23 guanines are contained in the aptamer G48nop sequence, which accounted for ~44% of the total number of bases (52 nt). The tendency of G48nop to form a G-quadruplex structure was also analyzed by the online QGRS Mapper (http://bioinformatics.ramapo.edu/QGRS/index.php (accessed on 10 February 2022)). The results showed that G48nop could form G-quadruplex and the sequence (5’-GGACGGGAGGTTGG-3’) inset in the G48nop sequence was of the highest G-score (Appendix A). It was reported that G-quadruplex DNA or RNA could fold into a stable structure and bind to various targets in vivo or in vitro [71,72]. Some aptamers with G-quadruplex motifs were also reported to bind to small molecules with high affinity, such as GTX [58], aflatoxin [73], lysine, arginine [74], etc.

The actual secondary structure of G48nop was measured by circular dichroism (CD) assays (Figure 4). The spectrum of the aptamer showed a positive peak at 216 nm, a broad positive band spanned from 255 to 306 nm, and a negative band spanned from 231 to 254 nm. This result implies that this aptamer contains a parallel G-quadruplex based on the 216 and 273 nm peak [75,76,77]. The addition of GYM-A induced a weak shift with a positive peak at 219 nm, a broad positive band spanned from 252 to 308 nm, and a negative band spanned from 240 and 251 nm. The increase in the positive 216 and 273 nm peak indicates the conformation change in the parallel G-quadruplex by the binding of the target GYM-A. As a control, the spectrum of the solution containing only GYM-A was relatively flat as GYM-A itself has no spectrum of circular dichroism.

### 2.4. Identification of Affinity and Specificity of G48nop

The affinity of G48nop was determined by different concentrations of GYM-A (1.75 μM, 3.5 μM, and 7 μM). The *K*_on_ and *K*_dis_ value of G48nop were 9.57 × 10^6^ M^−1^s^−1^ and 9.12 × 10^−1^ s^−1^, respectively, and the *K*_D_ value was 95.30 nM. Combined with the interaction curve (Figure 5A), a fast on-rate and fast off-rate interaction mode may be deduced for GYM-A and G48nop.

The specificity of G48nop was also determined by the BLI technique. According to chemical structures, marine biotoxins can be divided into the following four categories: polypeptide toxins, polyether toxins, alkaloid toxins, and cyclic imine (CI) toxins [29,78]. The chemical structures of the toxins in the first three categories are quite different from that of GYM-A, while the other members of the CI toxins have a high degree of structural similarity with GYM-A, due to the common structure of the spirocyclic imine ring. Thus, ten typical marine biotoxins from four categories were chosen and tested by dissolving in the same selection buffer of GYM-A. The representative marine biotoxins included the following: OA, BTX, dinophysistoxin (DTX), and PLTX from polyether toxins; GTX and STX from alkaloid toxins; MC-LR and nodularin-R (NOD-R) from peptide toxins; spirolide (SPX) and pinnatoxin (PnTX) from CI toxins. The blank samples contained only the selection buffer without any toxins. A random sequence fixed on the biosensor was used as an aptamer control. The results showed that G48nop had affinity only for samples containing GYM-A alone or GYM-A mixed with other toxins (the response values were 0.19 nm for GYM-A alone and 0.18 nm for the mixture) while this aptamer had no affinity for samples without GYM-A. Besides, the random sequence showed no binding to GYM-A (Figure 5B). These results indicate that G48nop can bind to GYM-A with high specificity, and this binding will not be interfered with by a variety of other marine biotoxins that may be present in high concentrations in complex shellfish samples [79,80,81,82].

### 2.5. Microscale Thermophoresis (MST)

The affinity and specificity of aptamer G48nop was further confirmed by MST assays. In MST, molecular mobility in the microscopic thermal gradient changes as the molecule changes in size, charge, or hydration. Thus, this change caused by thermophoresis can be recorded and used to measure the biomolecular interactions [83,84]. The results showed that, with the increasing concentration of GYM-A, the aptamer binding fraction curve showed as a sigmoidal curve. The determined *K*_D_ value of G48nop to GYM-A (34.50 ± 1.72 nM) was about one third of the *K*_D_ value (95.30 nM) determined by BLI (Figure 6). We speculated that, since G48nop was immobilized onto biosensors in the BLI assay, the folding and crimping of this aptamer might be limited when interacting with GYM-A, and the measured affinity decreased accordingly. The random sequence used as the control showed no binding with GYM-A (Figure 6) as the *K*_D_ value could not be fitted by the analysis system. In addition, G48nop showed no or only very low affinity to the four typical marine biotoxins including OA, STX, NOD-R, and PnTX chosen from the four categories, which further confirmed the specificity of this aptamer (Appendix A, Appendix A).

### 2.6. Label-Free BLI-Based Aptasenor for Detecting GYM-A

BLI is a label-free technology for measuring the interaction between biomolecules by an optical analytical system. Due to its advantages of rapidness, real-time, and low sample consumption, BLI has been gradually used in many areas [85,86]. Based on the DNA aptamer G48nop, which has high affinity and specificity, we developed an aptamer biosensor by BLI technology. The BLI technique is an efficient means to measure the biomolecular interactions. As shown in Figure 7A, white light emitted by the BLI spectrometer is reflected by two surfaces of the biosensor tip biological layer, and the reflected light generated from the two surfaces form an interferometric wave. When the affinity substance is attached to the tip surface, the thickness of the tip biological layer is altered, and the interferometric wave patterns is altered, which is displayed as the obvious spectral shift (Δ*λ*) of the biological layer as a function of time [86]. The performance of the aptasensor was assessed in order to verify its feasibility. In general, the linear range, limit of detection (LOD), limit of quantitation (LOQ), specificity, precision, and feasibility in real samples of the aptasensor were tested [87].

Firstly, the linear range, LOD, and LOQ of the aptasensor were measured by testing GYM-A samples in the concentration range of 55–14,000 nM. The results showed that the response values increased with the increasing GYM-A concentration (Figure 7B). The relationship between the response value and the concentration of GYM-A is illustrated in Figure 7C. A sigmoidal logistic five-parameter equation can be used to fit the curve, as follows:*y* = (R_max_ − R_min_)/[(1 + (*x*/EC_50_) ^b^)] + R_min_

In this equation, R_max_ is the maximum response value, R_min_ is the minimum response value, EC_50_ represents the concentration at which GYM-A reaches half of the maximum response value, and b is the slope of the curve. Then, we analyzed the data by Graphpad Prism 6.0 software (GraphPad Software Inc., San Diego, CA, USA) and obtained the following equation:*y* = (0.74030 − 0.06985)/[(1 + (*x*/3316)^−0.7837^)] + 0.06985

The correlation coefficient R^2^ of the data was analyzed to be 0.9963. The curve showed an excellent linear detection at 55–875 nM (Figure 7D), and a linear regression equation (*y* = 0.00018*x* + 0.09403) was fitted with a correlation coefficient R^2^ of 0.9916. The limit of detection (LOD) was 6.21 nM (calculated by 3S_a_/b) and the limit of quantification (LOQ) was 20.72 nM (calculated by 10S_a_/b) [88]. S_a_ was the standard deviation of the response of the blank samples (*n* = 20) and b was the slope of the calibration curve. These results show that this aptasensor has a high sensitivity in detecting GYM-A.

Secondly, the specificity of this aptasensor was assessed by testing eleven different kinds of toxins (GYM-A, OA, BTX, DTX, PLTX, GTX, STX, MC-LR, NOD-R, SPX, and PnTX) chosen from four categories of marine biotoxins. A mixture of these toxins was also tested by this aptasensor (Figure 7E). The concentration of each toxin (alone or in combination) was 1.5 μM. Each sample was tested three times. The samples containing GYM-A had significantly higher response values (0.29 nm of GYM-A alone and 0.28 nm of the mixture) than the samples without GYM-A. These results indicate that G48nop can bind to GYM-A with high specificity, and this binding will not be interfered with by a variety of other marine biotoxins that may be present in high concentrations in complex shellfish samples [79,80,81,82].

Thirdly, the precision of the aptasensor was assessed by repeating the test three times for GYM-A in a range of 55–14,000 nM. The relative standard deviations (RSDs) of inter- and intra-assay ranged from 0.034 to 4.57%, which were all below 5% (Appendix A). These results confirm the excellent precision of the aptasensor [89].

Fourthly, the feasibility of this aptasensor in real samples was also measured. (Table 3). GYM-A was spiked into water, or shellfish samples, at final concentrations of 875, 1750, and 3500 nM. These samples were tested for ten repetitions by this aptasensor. The recovery rates ranged from 96.65 to 109.67%, indicating that the sample matrix might not significantly interfere with the detection of GYM-A. Therefore, this aptasensor can be used in real samples. The RSDs obtained from three repeated experiments were as low as 0.55–2.28%, which also confirmed the accuracy of this aptasensor in real samples.

The concentration of GYMs in shellfish samples has been reported in several representative studies [1,3,81,90,91]. For example, in the HPLC quantitative detection and analysis study of GYM-A in Tunisia coastline polluted shellfish from 2000 to 2007, the average content in the lowest year reached 460 μg/kg and the highest year was 1290 μg/kg [90]. According to the method used in this study to extract GYM-A from shellfish samples, the LOD, LOQ, and the linear detection range for detecting GYM-A in shellfish samples can be calculated into 0.1575 μg/kg, 0.5255 μg/kg, and 1.3949–22.1920 μg/kg, respectively. Thus, according to these representative studies, the concentrations of GYM-A in most shellfish samples where GYMs have been detected were significantly higher than the LOD, LOQ, and the lower limit of the linear detection range of this aptasensor. Therefore, the aptasensor in this study is suitable for GYM-A detection in real shellfish samples. However, the concentration of GYM-A in seawater was generally very low. The average GYMs concentrations were reported to be only 0.0003 μg/L (bay) and 0.00003 μg/L (beach) in North Stradbroke Island, Queensland, Australia [1]. According to the extraction method of GYM-A in water samples adopted in this study, the LOD, LOQ, and the linear detection range for water samples can be calculated into 0.7875 μg/L, 2.6275 μg/L, and 6.9746–110.9601 μg/L, respectively. Therefore, this sensor may not be suitable for detection of seawater with very low concentrations of GYM-A.

Currently, the detection methods for GYMs are mainly based on LC–MS/MS or RBA (Table 4). Although they have certain advantages, such as low LOD or wide linear range, requirement of professional operators (LC–MS/MS), difficulty in obtaining experimental materials (RBA), or poor specificity (RBA) are the limitations of these methods in practice [92]. Compared with the RBA methods, the BLI-based aptasensor has high specificity and does not require the acquisition of precious experimental materials. Compared with LC–MS/MS methods, the BLI technology is simpler to operate [86]. By setting up the program, BLI instrument can perform fully automatic testing of large numbers of samples in a short period of time [86,93,94]. In addition, the biosensor detection tips directly interact with the samples, eliminating the need for microfluidics [86,93]. This not only simplifies the preparation process of the samples, but also helps to maintain the integrity of the samples for reuse.

In this research, the biorecognition element of GYM-A was the inexpensive DNA aptamer, which is easy to synthesize, store, transport, and is stable even in extreme temperature conditions. The BLI-based aptasensor enables the simultaneous detection of seven GYM-A samples in a small volume of 200 μL within 10 min. Apart from the BLI instrument, only the special biosensor tips and common 96-well plates were the main required materials, and the aptamer modified tips could be reused at least twenty times after washing for ~60 s in detection assays (Appendix A). In conclusion, the BLI-based aptasensor not only has the characteristics of high specificity, sensitivity, precision, and feasibility in real samples, but also is a high-throughput, low-cost, and convenient tool for the detection for GYM-A.

## 3. Conclusions

So far, there are no reports of aptamers targeting GYM-A or its analogues. In this study, high-affinity aptamers of the challenging small molecule GYM-A were successfully screened by Capture-SELEX. After further optimization, the aptamer G48nop with the highest *K*_D_ value of 95.30 nM was obtained. This aptamer showed high specificity for GYM-A, as determined by BLI and MST assays. G48nop was further used to establish a BLI-based aptasensor. This aptasensor exhibited high sensitivity, specificity, and precision in detecting GYM-A. Furthermore, the good recovery rate and repeatability of this aptasensor in detecting GYM-A in water and shellfish samples further confirmed its feasibility in real samples. Although this aptasensor is not suitable for the detection of seawater samples with very low GYM-A concentrations, its sensitivity is sufficient for most shellfish samples. In conclusion, this BLI-based aptasensor established in this work may be used as a sensitive and reliable tool to detect GYM-A, and the obtained aptamer G48nop with high affinity and specificity may be further used to develop into other types of aptasensors to facilitate the detection of GYM-A in the future.

## 4. Materials and Methods

### 4.1. Materials and Reagents

All ssDNA oligonucleotides were synthesized and purified by high-performance liquid chromatography (HPLC) by Sangon Biotechnology Co., Ltd. (Shanghai, China). Gymnodimine-A (GYM-A), brevetoxin (BTX), spirolide (SPX), and pinnatoxin (PnTX) were purchased from the National Research Council Canada (Halifax, NS, Canada). Okadaic acid (OA), dinophysistoxin (DTX), palytoxin (PLTX), gonyautoxin (GTX), saxitoxin (STX), microcystin-LR (MC-LR), and nodularin-R (NOD-R) were purchased from Taiwan Algal Science Inc. (Taiwan, China). Dynabeads^TM^ M-270 streptavidin (diameter 2.8 µm) and the Qubit^®^ ssDNA assay kits were purchased from Thermo Fisher Scientic (Chelmsford, MA, USA). GoTaqHot^®^ Start Colorless Master Mix was purchased from Promega Corporation (Fitchburg, WI, USA). QIAEX^®^ II gel extraction kits were purchased from Qiagen (Frankfurt, Germany). Super streptavidin (SSA) sensor biosensors were purchased from ForteBio (Shanghai, China). Selection Buffer (20 mM pH 7.6 Tris-HCl, 100 mm NaCl, 2 mM MgCl_2_, 5 mM KCl, and 1 mM CaCl_2_) was purchased from Tiandz (Beijing, China). Selection buffer was also used in biolayer interferometry (BLI) experiments. All reagents were of analytical grade and were used without further purification or treatment unless specified. All solutions were prepared with ultrapure water.

### 4.2. Instruments

The PCR assays were conducted by a PCR instrument (Bioer Technology, Hangzhou, China). The rotation of tubes was conducted by a four-dimensional rotating mixer (Kylin-Bell Lab Instruments Co., Ltd., Haimen, China). The DNA concentration was quantified by a Qubit^®^ 2.0 fluorometer (Life Technologies, Carlsbad, CA, USA). The Milli-Q water purification system was purchased form Millipore Corp (Bedford, MA, USA). BLI assays were conducted by a biolayer interferometry instrument (ForteBio, Shanghai, China). MST assays were conducted by a Monolith NT.115 system (NanoTemper Technologies, München, Germany). CD assays were tested by a Chirascan spectrometer (Applied Photophysics, Cardiff, UK.).

### 4.3. Capture-SELEX Library and Primers

A random Capture-SELEX library named Bank1 (5′-GGGAGGACGAAGCGGAAC-N_10_-TGAGGCTCGATC-N_30_-CAGAAGACACGCCCGACA-3′) was chemically synthesized. The oligonucleotides (88 nt) of this library included constant primer binding sites (18 nt each) flanking both ends, two different random sequences (N_10_ = 10 nt and N_30_ = 30 nt), and a specific docking sequence (12 nt) in the middle. Primer binding sites were designed for the binding of corresponding primers. The forward primer F1 was 5′-GGGAGGACGAAGCGGAAC-3′ and the reverse primer LR1 containing a poly adenine tail (A_20_ = 20 nt A) was 5′-A_20_-Spacer18-TGTCGGGCGTGTCTTCTG-3′. The modified reverse primer facilitated the separation of the two different-sized ssDNA sequences of PCR products with double strands. The forward primer and an unmodified reverse primer R1 (5′-TGTCGGGCGTGTCTTCTG-3′) were used to amplify the last round library for cloning. The docking sequence in the library enabled the complementary binding of the library to capture oligo (5′-biotin-GTC-Spacer18-GATCGAGCCTCA-3′), which was modified at 5′ end for binding to streptavidin beads.

### 4.4. Hybridization of the Library and Capture Oligo and Immobilization of Hybrid on Beads

In each round of selection, the library and capture oligo were dissolved and mixed in selection buffer at a molar ratio of 1:3. The mixture was heated to 95 °C for 10 min, cooled to 60 °C for 1 min, and cooled slowly to 25 °C at a rate of 0.1 °C per second (the hybridization was performed by a PCR instrument). The hybrid was then incubated with streptavidin beads (washed three times in 1000 μL of selection buffer before use) overnight on a four-dimensional rotating mixer.

### 4.5. Aptamer Selection by Capture-SELEX

After overnight incubation, the beads were collected by magnetic separation and washed six times in 1000 μL of selection buffer to remove unbound sequences. After that, the beads were incubated with 200 pmol of GYM-A (dissolved in 1000 μL of selection buffer) for 2 h on a four-dimensional rotating mixer at room temperature. The eluted ssDNA obtained by magnetic separation was quantified by a fluorometer and amplified by PCR (95 °C for 5 min; 20 reaction cycles containing 95 °C for 30 s, 56 °C for 30 s, 72 °C for 30 s; 72 °C for 5 min). Then, the PCR products were separated by 9% denaturizing urea-polyacrylamide gel electrophoresis (urea-PAGE). The small-sized ssDNA as the target fragment was purified by gel extraction, quantified, and used as the secondary library in the next round.

The input amount of ssDNA library was 2 nmol for round 1, 300 pmol for round 2–10, and 200 pmol for rounds 11 and 12. The input amount of GYM-A was kept at 200 pmol each round. During the selection, the pressure increased by adding the counter target of MC-LR, increasing the amount and incubation time of MC-LR, and reducing the incubation time of GYM-A. To improve the specificity, MC-LR was employed as a counter target in round 6–12. After six times of washing, the beads were incubated with MC-LR in 1000 μL of selection buffer. Then they were washed three times in 1000 μL of selection buffer before being incubated with the positive target GYM-A. Detail conditions of the selection are summarized in Appendix A.

### 4.6. Cloning, Sequencing, and Sequence Analysis

The ssDNA purified in the last round was amplified by primer F1 and R1. Cloning and sequencing of double-stranded PCR products were conducted by Sangon Biotechnology Co., Ltd. A total of 80 sequences in the sequencing results were clustered by Clustal X 2.1 software and divided into several families. The secondary structure and corresponding Gibbs free energy of candidate aptamers were predicted by the web server of mfold (http://www.unafold.org/mfold/applications/dna-folding-form.php (accessed on 10 February 2022)). The folding temperature was set to 25 °C, and the concentrations of Na^+^ and Mg^2+^ were set to 100 mM and 2 mM, respectively. Other parameters were set to be the default in mfold.

### 4.7. BLI Assays

BLI assays were applied to determine the binding affinity and specificity of aptamers or the established aptasensor. The detailed principle and operating procedure were followed from the report by Concepcion et al. (Appendix A) [86]. In this study, super streptavidin- (SSA-) coated biosensor tips were applied for the immobilization of biotin modified aptamers, and the tips were activated in selection buffer for 20 min before use. The aptamers were denatured at 95 °C for 10 min, quickly removed to 0 °C for 5 min, and were kept at room temperature for at least 5 min. The following five steps were contained in BLI assays: baseline 1 (60 s), loading (180 s), baseline 2 (60 s), association (90 s), and dissociation (60 s). The solutions of the baseline 1, baseline 2, and dissociation steps were the selection buffer. The solution of the loading step was the selection buffer containing 2 μM of biotin-modified aptamers. The solution of the association step was the selection buffer, in which designed concentrations of toxins were dissolved. The required solutions were all added to appropriate locations in a 96-well plate. To exclude the effect caused by the buffer, and nonspecific binding caused by the binding of toxin GYM-A and tips, the control wells containing only selection buffer without toxin GYM-A and the control SSA tips without the immobilization of aptamers were set. The ForteBio Data Analysis 9.0 Software (ForteBio, Shanghai, China) 21 CFR Part 11 version was used to analyze the data. The determined affinity constant *K*_D_, association constant *K*_on_, and dissociation constant *K*_dis_ were fused with a 1:1 binding mode.

### 4.8. MST Assays

MST assays were applied to confirm the binding affinity and specificity of aptamers and GYM-A. A Monolith NT.115 system was set with 15% excitation power and 40% MST power at 25 °C. The laser off/on times were 5 and 30 s, respectively. Carboxyfluorescein- (FAM-) labeled aptamers and unlabeled GYM-A were prepared in selection buffer supplemented with Tween-20 of a final concentration of 0.02%. Aptamers (200 nM) treated as the procedure in BLI assays were mixed with a titration series of GYM-A in equal volume, and the mixtures were loaded in capillaries. The final concentrations of GYM-A ranged from 0.305 to 10,000 nM. Assays were conducted more than three times and the affinity constant *K*_D_ was obtained from the MO. Affinity Analysis v2.1.2 software (NanoTemper Technologies, München, Germany).

### 4.9. CD Assays

CD assays were applied to determine the conformation of aptamers. A Chirascan spectrometer was applied with 0.05 cm path length quartz cuvettes. The scanning spectrum ranged from 210 to 320 nm with the step size of 1 nm and the integration time of 0.5 s. Each spectrum scan was the average of three duplicates at 20 °C. The molar ellipticity value was excluded and the selection buffer spectrum value was obtained under the same conditions. CD spectra was analyzed by the Pro-Data viewer software. Unlabeled aptamers (5 μM) were prepared in selection buffer and treated as the procedure in BLI assays before testing.

### 4.10. Treatment of Real Samples

Water samples were treated as in the report by Gao et al. [106]. Water samples (2 mL) spiked with GYM-A were centrifuged at a speed of 5000 rpm for 10 min. The obtained supernatant was added into a speedVac concentrator for evaporation. The residue was dissolved in 10 µL methanol, and then 10 µL methanol containing GYM-A was dissolved in 500 µL selection buffer. The solution was filtered through a 0.45 µm filter.

Shellfish samples were purchased from a market and were treated as in the report by Mukherjee et al. [107]. The GYM-A content of these samples was first determined. We only used those samples without GYM-A in the spiking experiment. GYM-A was spiked to 10 g shellfish homogenate and kept for 2 h at room temperature. A total of 30 mL of methanol was added to the homogenate and shaken for 1 min on a vortex mixer. After mixing, the samples were centrifuged at 3000 rpm for 10 min at 4 °C, and the supernatant was collected. Then, the homogenate was extracted one more time with the same procedure. The supernatants were pooled and concentrated, and the residue was resuspended in 60 mL ultrapure water. The dissolved solution was partitioned twice with 60 mL of CH_2_Cl_2_, and the CH_2_Cl_2_ was evaporated. The obtained residue was dissolved in 10 µL methanol, and then 10 µL methanol containing GYM-A was dissolved in 500 µL selection buffer. The solution was filtered through a 0.45 µm filter.

### 4.11. Statistical Analysis

Statistical analysis was performed by GraphPad Prism 6.0 Software (GraphPad Software Inc., San Diego, CA, USA), and the experiment data were shown as the mean ± standard deviation. One way ANOVA was used to analyze the variance and Dunnett’s *t*-test was used to compare the significance between all of the groups for the various experiments. The probabilities were *p* < 0.05 (*), *p* < 0.01 (**), *p* < 0.001 (***), and *p* < 0.0001 (****). The samples were tested at least three times per experiment.

## Figures and Tables

**Figure 1 toxins-14-00195-f001:**
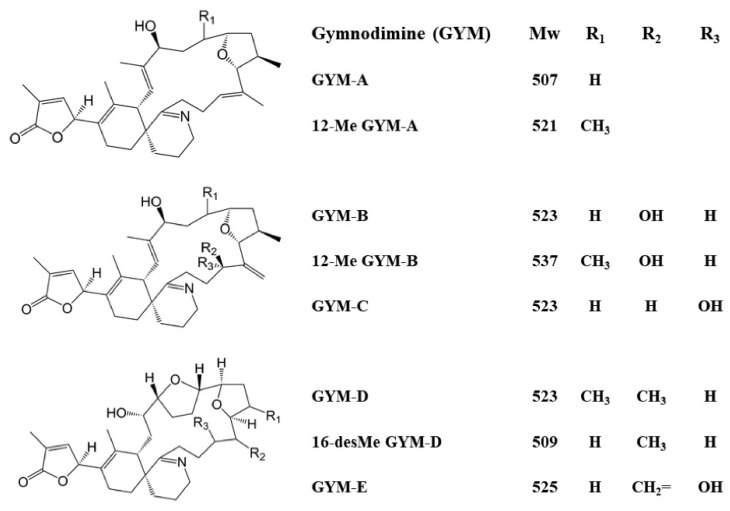
Chemical structures of GYMs and their congeners [6].

**Figure 2 toxins-14-00195-f002:**
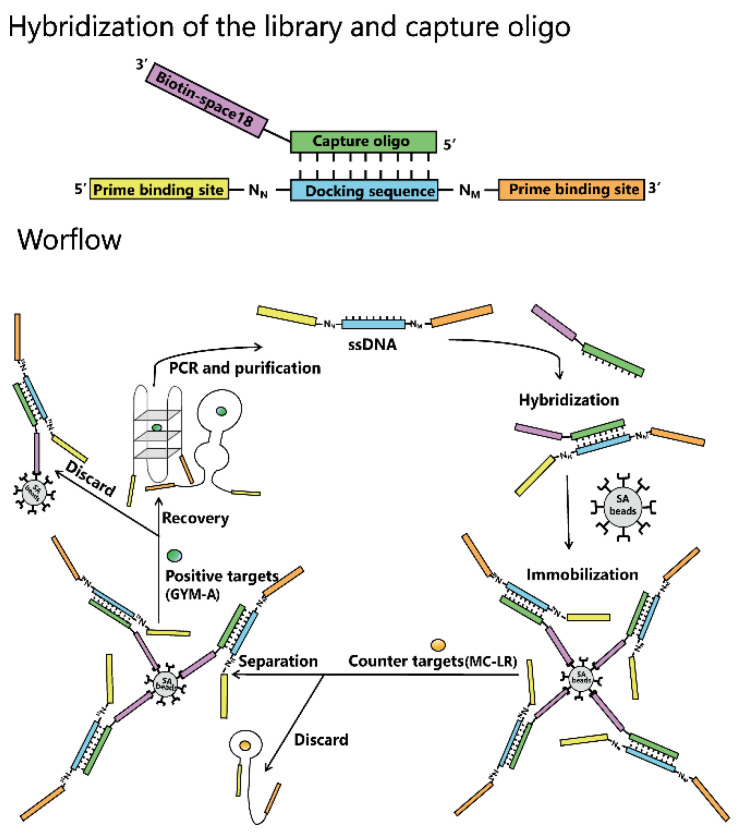
A general workflow of Capture-SELEX. Each round of Capture-SELEX mainly consists of four steps: (1) hybridization of the library and capture oligo by base pairing; (2) immobilization of the hybrid on beads (gray) via the strong interaction between streptavidin (on beads) and biotin (on capture oligos); (3) incubation with the positive target and elution of ssDNA (before being incubated with the positive target, the beads are washed several times with the selection buffer to remove ssDNA that are less strongly bound to the beads and incubated with the negative target to remove those ssDNA, which can bind to the negative target, thereby improving specificity); (4) PCR amplification of the eluted ssDNA and preparation ssDNA from these PCR products for the selection of next round.

**Figure 3 toxins-14-00195-f003:**
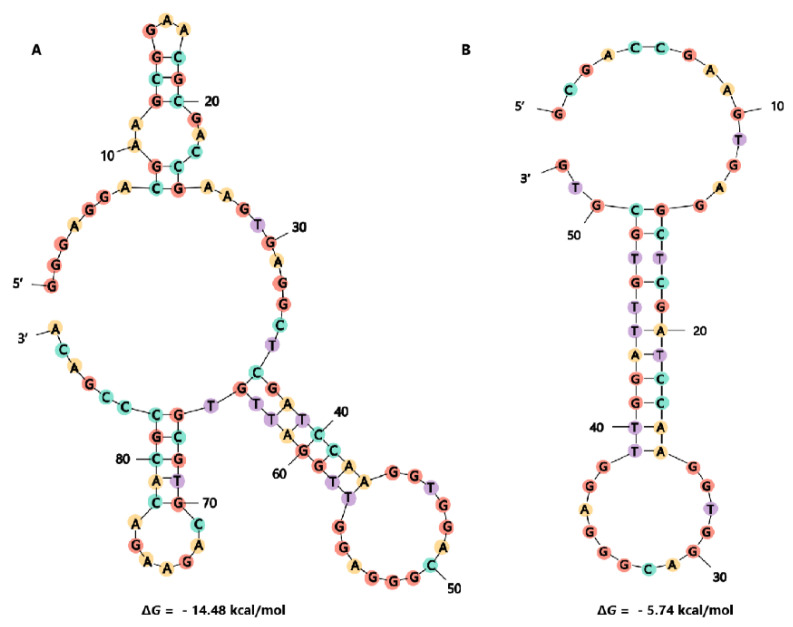
Secondary structure prediction of aptamer G48 (**A**) and aptamer G48nop (**B**) with their respective lowest Gibbs free energy value by mfold program. The folding temperature is 25 °C, and the concentrations of Na^+^ and Mg^2+^ were 100 mM and 2 mM, respectively.

**Figure 4 toxins-14-00195-f004:**
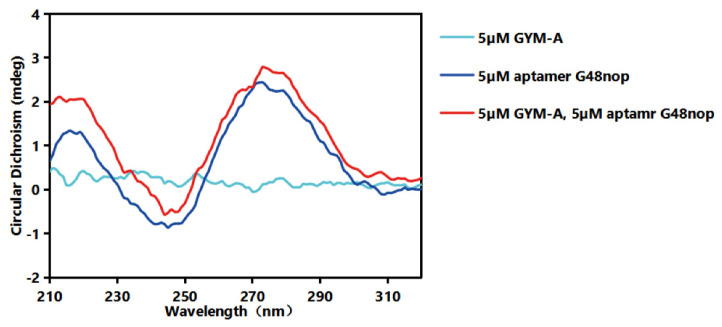
CD assays of aptamer G48nop and GYM-A samples. The cyan line stands for the CD spectra of GYM-A, the blue line stands for the CD spectra of aptamer G48nop, and the red line stands for the CD spectra of GYM-A and aptamer G48nop. The sample buffer is 20 mM Tris-HCl (pH 7.6), 100 mm NaCl, 2 mM MgCl_2_, 5 mM KCl, and 1 mM CaCl_2_.

**Figure 5 toxins-14-00195-f005:**
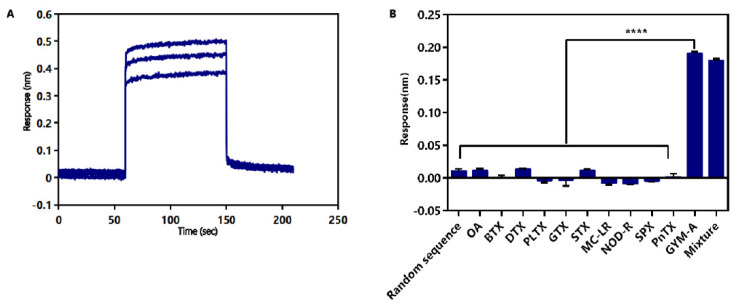
Identification of affinity and specificity of G48nop for GYM-A. (**A**) Identification of affinity of G48nop for GYM-A. The blue lines stand for the spectral shift of aptamer G48nop with GYM-A (7 μM (top), 3.5 μM (middle), and 1.75 μM (bottom)). (**B**) Identification of specificity of G48nop for GYM-A. The specificity of G48nop measured by BLI for eleven kinds of marine biotoxins (GYM-A, OA, BTX, DTX, PLTX, GTX, STX, MC-LR, NOD-R, SPX, and PnTX) and mixture is shown as response values caused by spectral shift. A random sequence fixed on the biosensor was used as an aptamer control. The final concentration of each marine biotoxin is 0.5 µM and the final concentration of each marine biotoxin in the mixture is also 0.5 µM. Every sample was tested three times. **** *p* < 0.0001 vs. GYM-A.

**Figure 6 toxins-14-00195-f006:**
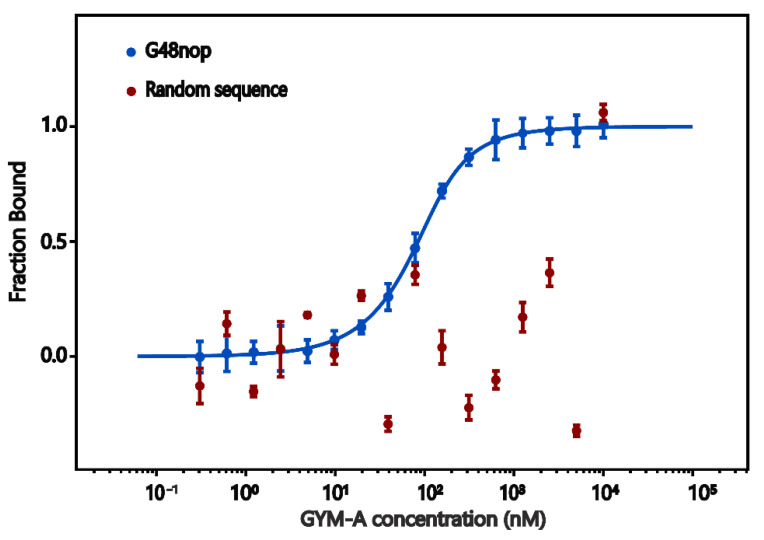
MST assays for aptamer G48nop (blue) and control sequence (red). The *K*_D_ value of G48nop is 34.50 ± 1.72 nM. The control sequence shows no binding affinity for GYM-A.

**Figure 7 toxins-14-00195-f007:**
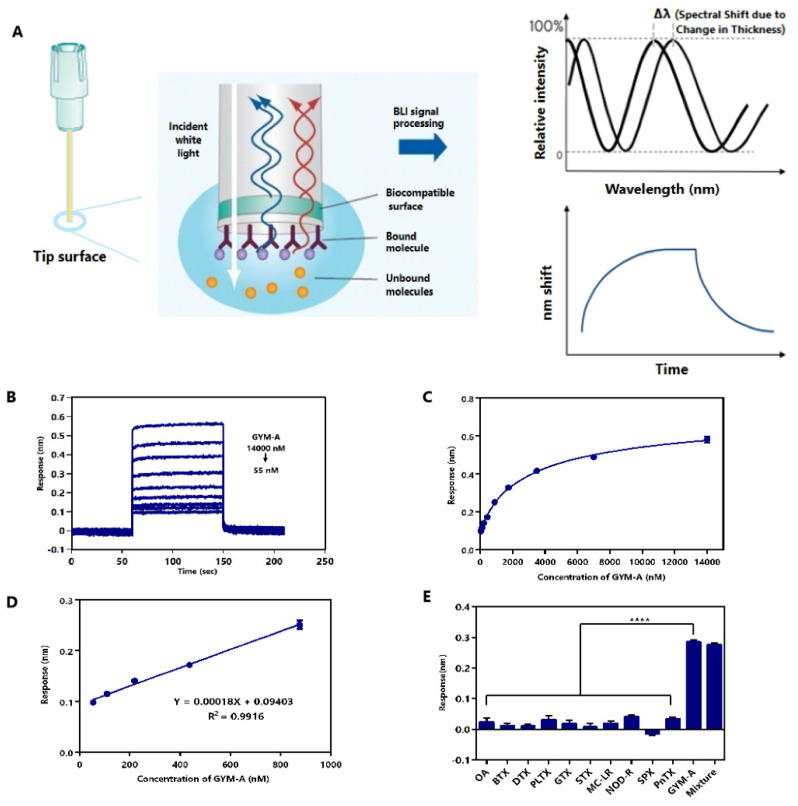
Evaluation of the performance of the BLI-based aptasensor. (**A**) The principle of BLI assay for detection [85,86]. (**B**) Response characterization of BLI-based biosensor measuring various concentrations (55–14,000 nM) of GYM-A. (**C**) Calibration curve of response values with the changing of various concentrations (55–14,000 nM) of GYM-A. The error bar stands for standard deviation. (**D**) Linear range of the calibration curve of GYM-A. A plot of response values with the changing of various concentrations (55–875 nM) of GYM-A. (**E**) Specificity of the BLI-based biosensor with eleven different kinds of toxins (GYM-A, OA, BTX, DTX, PLTX, GTX, STX, MC-LR, NOD-R, SPX, and PnTX, each at 1.5 µM) and the mixture of those toxins (each toxin in mixture was of a concentration of 1.5 µM). Every sample was tested three times. **** *p* < 0.0001 vs. GYM-A.

**Table 1 toxins-14-00195-t001:** Affinity constants (*K*_D_) of GYM-A candidate aptamers.

Name of DNA Aptamers	Family	Sequence (5′-3′)	*K*D (nM)
G48	A	GGGAGGACGAAGCGGAACGCGACCGAAGTGAGGCTCGATCCAAGGTGGACGGGAGGTTGGATTGTGCGTGCAGAAGACACGCCCGACA	288
G13	B	GGGAGGACGAAGCGGAACCCATGGGGTGTGAGGCTCGATCAGGAGGTAAAAGCAGCGCGTCGTCAGTGTGCAGAAGACACGCCCGACA	168,000
G16	C	GGGAGGACGAAGCGGAACCACCTTGAGATGAGGCTCGATCACCGGTGGGTATAGTGGTCGCGATGTATACCAGAAGACACGCCCGACA	495
G79	D	GGGAGGACGAAGCGGAACGGGGGTGGGTTGAGGCTCGATCATGATAGCGAATTGGACACGATTCGTGTACCAGAAGACACGCCCGACA	341
G29	E	GGGAGGACGAAGCGGAACCGTAGGAAAGTGAGGCTCGATCTACCCTTGGATCGATTTGTCAGGTGGGGACCAGAAGACACGCCCGACA	308
G24	F	GGGAGGACGAAGCGGAACGGGCGGGATTTGAGGCTCGATCCGACTACGGATACGGCTTGCTTGTATCCGCCAGAAGACACGCCCGACA	859

**Table 2 toxins-14-00195-t002:** Affinity constants of truncated forms of G48.

Name of DNA Aptamers	Sequence (5′–3′)	*K*_D_ (nM)
G48nop	GCGACCGAAGTGAGGCTCGATCCAAGGTGGACGGGAGGTTGGATTGTGCGTG	95.30
G48nors	TGAGGCTCGATCCAAGGTGGACGGGAGGTTGGATTGTGCGTG	354.00
G48norsj	CGATCCAAGGTGGACGGGAGGTTGGATTG	501.00

**Table 3 toxins-14-00195-t003:** Detection of GYM-A in real samples by aptasensors.

Samples	Spiked GYM-A (nM)	Recovery Rate (%)	RSD (%)
Shellfish	875	96.65	2.28
	1750	104.98	0.91
	3500	106.06	0.55
Water	875	105.10	1.09
	1750	109.67	1.28
	3500	99.63	1.87

**Table 4 toxins-14-00195-t004:** Comparison of instrumental detection methods of GYM-A.

Analytical Techniques	LOD	LOQ	Linear Range	Recovery Rate	Reference
Fluorescence Polarization	80 nM	—	—	82.8–98.4%	[30]
Microplate-receptor binding assay	2 nM	20 nM	—	—	[31]
Chemiluminescence	154 ± 64.3 nM	—	—	—	[95]
HPLC–UV	5 ng/mL	8 ng/g	0.005–1 µg/mL	96%	[90]
HPLC–MS/MS	0.06 ng/mL	0.2 ng/mL	0.90–42.6 ng/mL	92.1 ± 2.1%	[38]
UPLC–MS/MS	100 pM or 0.10 pg	1 nM or 1.0 pg	1pM–1µM	—	[96]
Pipette tip solid-phase extraction UPLC–MS/MS	0.1 µg/kg	0.2 µg/kg	0.5–100 ng/mL	86–97.92%	[97]
SPE-HPLC–MS/MS	0.052 µg/kg	0.16 µg/kg	0.5–16.0 ng/mL	71%	[98]
IT-APCI-MS/MS	1.00 pg	—	—	—	[99]
DMSPE-LC–HRMS/MS	0.03 ng/L	0.1 ng/L	0.01–2 µg/L	—	[100]
LC–MS/MS multiresidue method	11 µg/kg	20 µg/kg	—	81–107%	[101]
Online TFC-LC–MS/MS	0.5 µg/kg	1.5 µg/kg	2.5–200 µg/kg	92.7–116.0%	[102]
LC–HRMS	0.6 ng/mL (µg/kg)	1.2 ng/mL (µg/kg)	—	—	[36]
UHPLC–MS/MS	Mussel 0.11 µg/kg, P oyster 0.07 µg/kg	Mussel 0.38 µg/kg, P oyster 0.25 µg/kg	0.4–40 µg/kg	Mussel 84–88%, P oyster 88–90%	[103]
On-line SPE-LC–MS/MS	0.003 ng/L	0.007 ng/L	0.06–31.25 ng/L	90–90.5%	[104]
DLLME-LC–QqQ-MS/MS	0.7 ng/L	2.3 ng/L	2.5–1000 ng/L	98–112%	[37]
LC–ESI-MS/MS	1 μg/kg	3 μg/kg	0.5–20 ng/mL	—	[105]
BLI	6.21 nM	20.72 nM	55–875 nM	97–110%	This study

HPLC–UV: High performance liquid chromatography–ultraviolet; HPLC–MS/MS: High-performance liquid chromatography–tandem mass spectrometry; UPLC–MS/MS: Ultra performance liquid chromatograph–tandem mass spectrometry; SPE-HPLC–MS/MS: Solid-phase extraction high-performance liquid chromatography–tandem mass spectrometry; IT-APCI-MS/MS: Ion trap-atmospheric pressure chemical ionization-mass spectrometry; DMSPE-LC–HRMS/MS: Dispersive micro solid-phase extraction–liquid chromatography–high-resolution mass spectrometry tandem mass spectrometry; LC–MS/MS: Liquid chromatography–tandem mass spectrometry; Online TFC-LC–MS/MS: Online turbulent flow chromatography coupled to liquid chromatography–tandem mass spectrometry; LC–HRMS: Liquid chromatography–high-resolution mass spectrometry; UHPLC–MS/MS: Ultra-high performance liquid chromatograph–tandem mass spectrometry; Online SPE-LC–MS/MS: Online solid phase extraction coupled to liquid chromatography–tandem mass spectrometry; DLLME-LC–QqQ-MS/MS: Dispersive liquid–liquid microextraction and liquid chromatography with triple quadrupole mass spectrometry; LC–ESI-MS/MS: Liquid chromatography–electrospray-ionization tandem mass spectrometry.

## Data Availability

The data presented in this study are available in this article and Appendix A.

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
