# Peer review of "Selection, Characterization, and Optimization of DNA Aptamers against Challenging Marine Biotoxin Gymnodimine-A for Biosensing Application"

_toxins, 2022, doi:10.3390/toxins14030195_

Round 1

Reviewer 1 Report

In the manuscript the authors focus on the selection of novel aptamer strand for detection of Gymnodimine-A (GYM-A). The conditions and the SELEX steps were presented along with the affinity studies. Moreover, the authors showed the studies on the BLI-based biosensor and presented its working parameters. I find the article interesting and I would have some more queations regarding the manuscript:

  • why did the authors choose the concentration of sodium and magnesium ions at 100 and 2 mM to evaluate the secondary structure of aptamer?
  • the authors show the linear range of response of BLI-based biosensor from 55 to 875 nM - I assume that for higher concentrations a plot flatenning was observed, but what was the response for concentrations below 55 nM?
  • why did the authors choose 1.5 uM concentration to test the selectivity of the sensor - should it be rather the concentration within linear range of biosensor response?
  • how many repetitions of precision test as well as real sample studies were conducted?

Author Response

Dear reviewer:

Sincere thanks should be given to the reviewer for the encouraging comments and constructive suggestions. The response to the comments is as follows. Thank you so much for your excellent comments and suggestions. We are very grateful to you for your valuable help and guidance, since we have leant a lot of knowledge from these important and constructive comments. We have tried to do our best to revise the manuscript accordingly. Please find the following detailed responses to your comments and suggestions.

With best regards.

Comments and Suggestions for Authors

In the manuscript the authors focus on the selection of novel aptamer strand for detection of Gymnodimine-A (GYM-A). The conditions and the SELEX steps were presented along with the affinity studies. Moreover, the authors showed the studies on the BLI-based biosensor and presented its working parameters. I find the article interesting and I would have some more queations regarding the manuscript:

Comments 1: why did the authors choose the concentration of sodium and magnesium ions at 100 and 2 mM to evaluate the secondary structure of aptamer?

Response: Thanks for your question. In the selection buffer, the concentrations of sodium and magnesium ions are 100mM and 2mM, respectively. Therefore, we used the same ionic conditions to evaluate the secondary structure of aptamers in mfold program.

Comments 2: the authors show the linear range of response of BLI-based biosensor from 55 to 875 nM - I assume that for higher concentrations a plot flatenning was observed, but what was the response for concentrations below 55 nM?

Response: Thanks for your question. In this work, we have tested different concentrations of GYM-A (including those below 55nM). However, when the concentrtions of GYM-A were below 55nM, the response values of this aptasensor did not fit the linear equation very well. This may be due to the fact that when the concentration of GYM-A is very low, its specific response is also very low, and therefore, the interference effect of non-specific response generated by solution components or non-specific binding of GYM-A is greatly enhanced.

Comments 3: why did the authors choose 1.5 uM concentration to test the selectivity of the sensor - should it be rather the concentration within linear range of biosensor response?

Response: Thanks for your question and kind suggestions. In this work, we have also chosen lower concentrations of these toxins to test the selectivity of this aptasensor. We found that with the increasing concentrations of these toxins, the nonspecific response resulting from nonspecific binding of these toxins was also increased. Therefore, testing the response of this aptasensor to GYM-A and other toxins at higher concentrations (beyond the linear detection range) and comparing them with each other benefit to elucidate the selectivity of this aptasensor.

Comments 4: how many repetitions of precision test as well as real sample studies were conducted?

Response: Thanks for your question. In the precision test, three repetitions were conducted. Please see in Line 332-333 in the revised manuscript. In real sample assay, ten repetitions were conducted. We have added relevant contents in the revised manuscript. Please see in Line 338 in the revised manuscript.

Reviewer 2 Report

toxins-1616959: Selection, Characterization and Optimization of DNA Ap-2 tamers against Challenging Marine Biotoxin Gymnodimine-A for Biosensing Application

The manuscript described the screening of aptamers targeting Gymnodimine-A using Capture-SELEX. The authors identified G48nop as the optimized aptamer with high specificity towards Gymnodimine-A. Subsequently, the authors established an aptasensor to detect Gymnodimine-A. Overall, the study is of high interest to the field. It was appropriately designed and implemented. However, there are several issues to consider before publishing this manuscript in Toxins. Please see the detailed comments as follows to revise the manuscript.

  1. The paragraph in lines 67-78 should be expanded (more details should be provided) to highlight the need for a novel detection method of GYM-A. Many LC-MS/MS methods are available (as shown in Table 4); they should be mentioned to provide a complete overview of the available detection methods for GYM-A.
  2. The paragraph in lines 79-85 should be expanded. The aptamer is the main focus of this study; therefore, the authors should provide a sufficient introduction of aptamer, its merits, and recent applications, particularly for detecting some small molecules like GYM-A.
  3. Line 105: After introducing Figure 2, it will be more informative if the authors briefly mention the primary steps of the workflow in this figure.
  4. Specificity of G48nop for GYM-A: the authors showed in Figures 5B and 7E that G48nop could bind to GYM-A with high specificity, and this binding was not interfered by a variety of other marine biotoxins present in complex environmental samples. However, the authors need to show the conventional concentration ranges of these biotoxins to strengthen the point. If one biotoxin is generally presented in environmental samples at 10-100 times higher than GYM-A, the interference will be considerable.
  5. How about the specificity of G48nop for other congeners of GYM-A, e.g., those presented in Figure 1? LC-MS/MS will be able to distinguish them. Will G48nop perform similarly?
  6. What is the conventional concentration range of GYM-A in marine samples? Please compare it with the linear range, LOD, and LOQ of the aptasensor and discuss.
  7. Was the aptasensor sensitive enough to replace LC-MS/MS in quantitation of GYM-A?
  8. Shellfish samples: did the authors determine the amount of GYM-A in the sample before spiking standard GYM-A?
  9. The authors should include a section in the Method to clarify the number of replications per experiment, how the data were presented (means ± SDs or sEs), and the statistical test.

Some minor points:

  1. Lines 54-56: uncomplete sentence.
  2. Figure 7D: the equation Y = aX + b should show the same decimal digits of a and b.

Author Response

Dear reviewer:

Sincere thanks should be given to the reviewer for the encouraging comments and constructive suggestions. The response to the comments is as follows. Thank you so much for your excellent comments and suggestions. We are very grateful to you for your valuable help and guidance, since we have leant a lot of knowledge from these important and constructive comments. We have tried to do our best to revise the manuscript accordingly. Please find the following detailed responses to your comments and suggestions.

With best regards.

Comments and Suggestions for Authors

The manuscript described the screening of aptamers targeting Gymnodimine-A using Capture-SELEX. The authors identified G48nop as the optimized aptamer with high specificity towards Gymnodimine-A. Subsequently, the authors established an aptasensor to detect Gymnodimine-A. Overall, the study is of high interest to the field. It was appropriately designed and implemented. However, there are several issues to consider before publishing this manuscript in Toxins. Please see the detailed comments as follows to revise the manuscript.

Comments 1: The paragraph in lines 67-78 should be expanded (more details should be provided) to highlight the need for a novel detection method of GYM-A. Many LC-MS/MS methods are available (as shown in Table 4); they should be mentioned to provide a complete overview of the available detection methods for GYM-A.

Response: Thanks for your kind suggestions. We are responsible for having not provide a complete overview of the available detection methods for GYM-A. In the revised manuscript, we expanded the corresponding content and please see Line 69-91 in the revised manuscript.

Comments 2: The paragraph in lines 79-85 should be expanded. The aptamer is the main focus of this study; therefore, the authors should provide a sufficient introduction of aptamer, its merits, and recent applications, particularly for detecting some small molecules like GYM-A.

Response: Thanks for your kind suggestions. The paragraph in lines 79-85 have been expanded to provide a more detailed introduction of aptamer (including its merits, recent applications and detectiong of marine tonxins like GYM-A) and please see Line 96-125 in the revised manuscript.

Comments 3: Line 105: After introducing Figure 2, it will be more informative if the authors briefly mention the primary steps of the workflow in this figure.

Response: Thanks for your kind suggestions. We have added the corresponding contents in the revised manuscript. Please see Line 159-167 in the revised manuscript.

Comments 4: Specificity of G48nop for GYM-A: the authors showed in Figures 5B and 7E that G48nop could bind to GYM-A with high specificity, and this binding was not interfered by a variety of other marine biotoxins present in complex environmental samples. However, the authors need to show the conventional concentration ranges of these biotoxins to strengthen the point. If one biotoxin is generally presented in environmental samples at 10-100 times higher than GYM-A, the interference will be considerable.

Response: Thanks for your kind suggestions, which is valuable for increasing the integrity of the manuscript. By reviewing relevant literatures, we found that the concentration range of these toxins for specific detection (including GYM, OA, DTX, STX, GTX, SPX, PnTX, etc.) varies greatly with regions, shellfish species, seasons, years, and even sampling depths. During the study of seasonal variation of algal toxins in North Stradbroke Island, Queensland, Australia, the average concentration of GYM was 8-15 μg/kg with a maximum of 137-220 μg/kg [1]. In the survey study of liposoluble toxins in the Chinese market, the concentration range of GYM was 0.10-14.4 μg/kg (the median concentration was about 0.78-1.42 μg/kg), while the concentration of OA in the same batch of samples ranged from 2.0-37.3 μg/ kg and the median concentration of OA was about 4-5 times that of GYM [2]. Concentrations of DSTs (including OA and DTX) above the guideline level of 160 μg/kg were found to be widespread in a 2012 surveillance of diarrheal shellfish toxins in Washington State [3]. In the research on paralytic shellfish toxins in Korea, the total content of five paralytic shellfish toxins (STX GTX1, GTX2, GTX3 and GTX4) in mussels averaged 101.4 μg/kg, and the maximum value reached 198.7 μg/kg [4]. We also reviewed the related literatures of SPX and PnTX, which belong to the cyclic imine family and have high structural similarity to GYM. In the French shellfish toxin monitoring report, the highest concentration of SPXs can reach 87 μg/kg [5]. In Rangaunu Harbour, New Zealand, the total PnTX content can be as high as 196.7 μg/kg [6]. In Norwegian blue mussels, the average value of PnTX G was 8.0 μg/kg, and the highest was 115 μg/kg. The total SPXs was up to 274 μg/kg [7]. Therefore, in a given sample, these toxins (including OA, DTX, STX, GTX, SPX, PnTx, etc.) may be present in much higher concentrations than GYM, which may interfere with the detection of GYM.

   According to reviewer's suggestion, we have revised and supplemented several references in the revised manuscript. Please see in Line 249-252 and 325-328 in the revised manuscript. Thanks again for your suggestion.

Comments 5: How about the specificity of G48nop for other congeners of GYM-A, e.g., those presented in Figure 1? LC-MS/MS will be able to distinguish them. Will G48nop perform similarly?

Response: Thanks for your question and kind suggestions, which is valuable for providing more comprehensive information on the specificity of G48nop. In this study, we tested the specificity of G48nop for eleven marine toxins (including SPX and PnTX, which belong to the cyclic imine family and are highly structurally similar to GYM-A). However, these results are not sufficient. To fully measure the specificity of G48nop, we should test the specificity of G48nop for other congeners of GYM-A.

However, GYM-A is the only congeners of GYMs for sale which was purchased from the National Research Council Canada (Halifax, Canada), and to our knowledge, no other congeners of GYMs are commercially available.

The Reviewer’s suggestion is very important. In future work, we will try to obtain GYM-A congeners by other means and test specificity of G48nop for them (for example, we can try to extract related toxins directly from shellfish). But these results cannot be presented in this article for the reasons of time. We will follow the reviewer's suggestion to better refine the evaluation of specificity in the future.

Comments 6: What is the conventional concentration range of GYM-A in marine samples? Please compare it with the linear range, LOD, and LOQ of the aptasensor and discuss.

Response: Thanks for your question and kind suggestion, which is valuable for increasing the integrity of these contents. According to several representative references [1, 2, 8-10], the detection concentration of GYM varies greatly with region, season, year, shellfish species, and tissue site. The minimum detection concentration is also affected by the detection method. In a historical survey of the accumulation of GYM concentrations in New Zealand shellfish samples, it was found that the GYM concentrations of shellfish samples ranged from 14.8 to 23437.0 μg/kg, with an average of 208.1 μg/kg (excluding the highest species), 3818.8 μg/kg (including highest species) [8]. In the HPLC quantitative detection and analysis study of GYM-A in Tunisia coastline polluted shellfish from 2000 to 2007, the average content in the lowest year reached 460 μg/kg and the highest year was 1290 μg/kg [9]. During the study of seasonal variation of algal toxins in North Stradbroke Island, Queensland, Australia, the average concentration of GYM was 8-15 μg/kg, and the maximum value was 137-220 μg/kg [1]. In the study on the distribution of lipid-soluble toxins in shellfish products in the Chinese seafood market [2], the concentration range of GYM was 0.1-14.4 μg/kg, and the median of the samples tested was 0.78-1.42 μg/kg. The LOD of the aptasensor obtained in this study is 6.21 nmol/L, the LOQ is 20.72 nmol/L, and the linear detection range is 55-857 nmol/L. According to the method used in this study to extract GYM-A from shellfish samples, the LOD, LOQ and the linear detection range can be calculated into 0.1575 μg/kg, 0.5255 μg/kg, 1.3949-22.1920 μg/kg, respectively. According to the concentration range of these representative reports. The concentrations of GYM in most shellfish samples were higher than the LOD, LOQ and the lower limit of the linear detection range of this aptasensor. Shellfish samples above the upper limit of the linear detection range can be quantified by dilution during the actual detection process. Therefore, the aptasensor in this study is suitable for GYM detection of shellfish samples. However, GYM concentrations in seawater are generally very low. The average GYM concentrations were only 0.00030 μg/L (bay) and 0.00003 μg/L (beach) in a study of algal toxins in North Stradbroke Island, Queensland, Australia [1]. According to the extraction method of GYM in water samples adopted in this study, the LOD, LOQ and the linear detection range can be converted into 0.7875 μg/L, 2.6275 μg/L and 6.9746-110.9601 μg/L, respectively. Therefore, this sensor may not be suitable for GYM detection in seawater.

According to your suggestion, we have added relevant descriptions and references in the revised manuscript. Please see in Line 344-362 in the revised manuscript. We hope that our modification can meet your requirements. Thanks again for your suggestion.

Comments 7: Was the aptasensor sensitive enough to replace LC-MS/MS in quantitation of GYM-A?

Response: Thanks for your question. Almost all LC-MS/MS based methods had lower LOD and LOB than BLI-based aptasensor, and their linear ranges were broader than the BLI. However, LC-MS/MS needs complex preprocessing of sample and special technical personnel. Compared with LC-MS/MS methods, the BLI technology is simpler to operate. By setting up the program, BLI can perform fully automatic testing of large numbers of samples in a short period of time. In addition, the biosensor detection tips directly interact with the samples, eliminating the need for microfluidics. This not only simplifies preparation process of the samples, but also helps to maintain the integrity of the samples for reuse. Thus, the BLI-based aptasensor has its own merits compared with classic LC-MS/MS methods. Therefore, we believe that under some specific conditions, BLI-based aptasensor can be an alternative to LC-MS/MS for high-throughput, rapid and low-cost detection of GYM-A.

Comments 8: Shellfish samples: did the authors determine the amount of GYM-A in the sample before spiking standard GYM-A?

Response: Thanks for your question. Before spiking standard GYM-A into shellfish samples, we would firstly determine whether or not the samples contained GYM-A. We only used shellfish samples without GYM-A. However, we are responsible for not making corresponding statements clear in the original manuscript. We have supplemented the corresponding statements in the revised manuscript. Please see Line 560-561 in the revised manuscript.

Comments 9: The authors should include a section in the Method to clarify the number of replications per experiment, how the data were presented (means ± SDs or sEs), and the statistical test.

Response: Thanks for your kind suggestions, we have added a section in the Method section. Please see Line 571-577 in the revised manuscript.

Comments 1 in some minor points: Lines 54-56: uncomplete sentence.

Response: Thanks for your kind suggestions. The sentence in Lines 54-56 in the original manuscript have been changed. Please see Lines 56-58 in the revised manuscript.

Comments 2 in some minor points: Figure 7D: the equation Y = aX + b should show the same decimal digits of a and b.

Response: Thanks for your kind suggestions. The equation Y = aX + b had changed into y = 0.00018x + 0.09403. Please see in Line 313 And Figure 7C in the revised manuscript.

Reference

  1. Takahashi, E.; Yu, Q.; Eaglesham, G.; Connell, D. W.; McBroom, J.; Costanzo, S.; Shaw, G. R., Occurrence and seasonal variations of algal toxins in water, phytoplankton and shellfish from North Stradbroke Island, Queensland, Australia. Marine Environmental Research 2007, 64, (4), 429-442.
  2. Haiyan; Wu; Jianhua; Yao; Mengmeng; Guo; Zhijun; Tan; Deqing; Zhou, Distribution of Marine Lipophilic Toxins in Shellfish Products Collected from the Chinese Market. Marine Drugs 2015, 13, (7), 4281-4295.
  3. Trainer, V.; Moore, L.; Bill, B.; Adams, N.; Harrington, N.; Borchert, J.; Da Silva, D.; Eberhart, B. T., Diarrhetic Shellfish Toxins and Other Lipophilic Toxins of Human Health Concern in Washington State. Marine Drugs 2013, 11, (6).
  4. Shin; Choonshik; Jo; Hyejin; Kim; Sheen-Hee; Kang; Gil-Jin, Exposure assessment to paralytic shellfish toxins through the shellfish consumption in Korea. Food Research International 2018.
  5. Zouher, A.; Manoella, S.; Florence, R.; Nadine, M.; Eric, A., Report on the First Detection of Pectenotoxin-2, Spirolide-A and Their Derivatives in French Shellfish. Marine Drugs 2007, 5, (4), 168-179.
  6. Mcnabb, P. S.; Mccoubrey, D. J.; Rhodes, L.; Smith, K.; Holland, P. T., New perspectives on biotoxin detection in Rangaunu Harbour, New Zealand arising from the discovery of pinnatoxins. Harmful Algae 2012, 13, (1), 34-39.
  7. Rundberget, T.; Aasen, J. A. B.; Selwood, A. I.; Miles, C. O., Pinnatoxins and spirolides in Norwegian blue mussels and seawater. Toxicon 2011, 58, (8), 700-711.
  8. Stirling, D. J., Survey of historical New Zealand shellfish samples for accumulation of gymnodimine. New Zealand Journal of Marine and Freshwater Research 2001, 35, (4), 851-857.
  9. Marrouchi, R.; Dziri, F.; Belayouni, N.; Hamza, A.; Benoit, E.; Molgó, J.; Kharrat, R., Quantitative Determination of Gymnodimine-A by High Performance Liquid Chromatography in Contaminated Clams from Tunisia Coastline. Marine Biotechnology 2010, 12, (5), 579-585.
  10. Krock, B.; Pitcher, G. C.; Ntuli, J.; Cembella, A. D., Confirmed identification of Gymnodimine in Oysters from the west coast of South Africa by Liquid Chromatography-Tandem Mass Spectrometry. South African Journal of Marine Science 2009, 31, (1), 113-118.

Round 2

Reviewer 2 Report

toxins-1616959: Selection, Characterization and Optimization of DNA Ap-2 tamers against Challenging Marine Biotoxin Gymnodimine-A for Biosensing Application

The authors appropriately revised and improved the manuscript based on previous comments. However, a minor revision is required. As the authors stated in the response letter, the study had some limitations. Therefore, these issues should be included in the Conclusion section.

Author Response

Dear reviewer:

Sincere thanks should be given to the reviewer for the encouraging comments and constructive suggestions. The response to the comments is as follows. Thank you so much for your excellent comments and suggestions. We are very grateful to you for your valuable help and guidance, since we have leant a lot of knowledge from these important and constructive comments. We have tried to do our best to revise the manuscript accordingly. Please find the following detailed responses to your comments and suggestions.

With best regards.

Comments and Suggestions for Authors

toxins-1616959: Selection, Characterization and Optimization of DNA Aptamers against Challenging Marine Biotoxin Gymnodimine-A for Biosensing Application

The authors appropriately revised and improved the manuscript based on previous comments. However, a minor revision is required. As the authors stated in the response letter, the study had some limitations. Therefore, these issues should be included in the Conclusion section.

Response: Thanks for your kind suggestions, which is valuable for increasing the integrity of this content. We have added a brief description, mainly concerning the limitations of this aptasensor in the detection of real samples (not suitable for detection of seawater samples with very low GYM-A concentration) in the revised manuscript. Please see in Line 421-423 in the revised manuscript (marked in green). Thank you again for your suggestion.
